# Copper–cobalt double metal cyanides as green catalysts for phosphoramidate synthesis

Alejandro Fonseca[1,2], Aram L. Bugaev [3,4], Anna Yu. Pnevskaya [3], Kwinten Janssens[1], Carlos Marquez [1✉] &
Dirk De Vos [1✉]

Phosphoramidates are common and widespread backbones of a great variety of fine chemicals, pharmaceuticals, additives and natural products. Conventional approaches to their synthesis make use of toxic chlorinated reagents and intermediates, which are sought to be avoided at an industrial scale. Here we report the coupling of phosphites and amines promoted by a $Cu_3[Co(CN)_6]_2$-based double metal cyanide heterogeneous catalyst using $I_2$ as additive for the synthesis of phosphoramidates. This strategy successfully provides an efficient, environmentally friendly alternative to the synthesis of these valuable compounds in high yields and it is, to the best of our knowledge, the first heterogeneous approach to this protocol. While the detailed study of the catalyst structure and of the metal centers by PXRD, FTIR, EXAFS and XANES revealed changes in their coordination environment, the catalyst maintained its high activity for at least 5 consecutive iterations of the reaction. Preliminary mechanism studies suggest that the reaction proceeds by a continuous change in the oxidation state of the Cu metal, induced by a $O_2/I^-$ redox cycle.

[1] Centre for Membrane Separations, Adsorption, Catalysis and Spectroscopy for Sustainable Solutions, KU LeuvenCelestijnenlaan 200F, 3001 Leuven, Belgium. [2] Department of Polymer Engineering and Science, Polymer Processing, Montanuniversitaet Leoben, Otto Gloeckel-Strasse 2, 8700 Leoben, Austria. [3] The Smart Materials Research Institute, Southern Federal University, Sladkova 178/24, Rostov-on-Don 344090, Russia. [4] Paul Scherrer Institute, Forschungsstrasse 111, 5232 Villigen, Switzerland. ✉email: carlos.marquez@kuleuven.be; dirk.devos@kuleuven.be

The phosphoramidate moiety is present in a wide variety of high-value natural and synthetic compounds, ranging from biologically active natural products, such as nucleotides[1], to ligands in metal catalysis and environmentally friendly flame retardant additives[2]. Such is the interest in these products that several synthetic methodologies to obtain them have been developed during the last decade[3–7].

Early strategies for phosphoramidate synthesis relied on the direct reaction of amines with the corresponding phosphoryl halides (Fig. 1a)[8,9]. These straightforward methodologies often use highly toxic reagents, such as the phosphoryl halide itself, and unpractical conditions, such as pre-cooling and fuming reactions[8,9]. An alternative to the direct use of halogenated phosphor substrates can be their in situ formation. For example, in the Atherton–Todd reaction a phosphoryl chloride is generated by the halogenation of phosphites using $CCl_4$ as a chlorine donor (Fig. 1b)[10]. However, the use of stoichiometric amounts of $CCl_4$ is highly undesired from an industrial point of view. More recent alternatives for phosphoramidate synthesis include the formation of phosphoryl azide precursors (Fig. 1c)[11], the direct electrochemical oxidation of phosphites and amines[12], the selenite-catalyzed Atherton–Todd-like reaction of phosphites and amines (Fig. 1d)[13] and the $ZnI_2$ triggered catalytic oxidative coupling of P(O)-H compounds and amines using an organic oxidant (Fig. 1e)[14]. While these methods are well-established, they all suffer from either dependency on non-desired precursors or poor atom/step economy.

A more environmentally friendly approach to the phosphoramidate synthesis is the use of metals for the aerobic oxidative coupling of phosphites and amines. In this sense, homogeneous Cu salts have shown very promising results[15,16]. Fraser et al.[17] proposed a mild and atom-efficient synthesis of phosphoramidates starting from phosphites and amines using CuI as catalyst; however, this approach suffered from the formation of oxidation side products[17]. Furthermore, similar homogeneously catalyzed aerobic oxidative cross-coupling systems for P-N bond formation have been investigated, with Cu emerging as the metal of choice for this transformation[18,19]. Inspired by these reports we envisioned a system that includes all the attractive features of the metal-catalyzed oxidative coupling, while also introducing the benefits of a heterogeneously catalyzed system.

Double metal cyanides (DMCs) are coordination polymers in which two different metals are linked through a cyanide group (C≡N). DMCs have the general formula $M^1_u[M^2(CN)_n]_v$ (sometimes expressed as $M^1$-$M^2$ DMC for simplicity). $M^1$ typically is a divalent metal ($v = 2$), like Zn(II) or Cu(II), and $M^2$ is a trivalent metal ($u = 3$), like Co(III) or Fe(III); however, other $u$-$v$ combinations are also possible, such as 1–1, 4–3[20]. Depending on the valences of the metal ions, $n$ can be 4, 6, or 8, with 6 being the most common for materials crystallizing in the cubic phase. The theoretical defect-free crystal structure of the Cu-Co DMC is shown in Fig. 1. However, due to the charge imbalance between $Cu^{2+}$ and $[Co(CN)_6]^{3-}$, one-third of all hexacyanocobaltates should be absent in the cubic lattice, creating vacancies in the coordination spheres of the $Cu^{2+}$ ions. Additionally, in practice, these DMCs are synthetized using different additives, as well as an excess of the $M^1$ source to increase their catalytic activity, which makes their true structures more complex[21–23]. This type of coordination polymer first found application as catalyst for the polymerization of epoxides[24–26], but to date, their catalytic capabilities remain relatively unknown by the broad scientific community. Nonetheless, the past decade has witnessed the emergence of new potential catalytic applications DMCs[27–33]. As oxidation catalysts, their potential has been explored in the

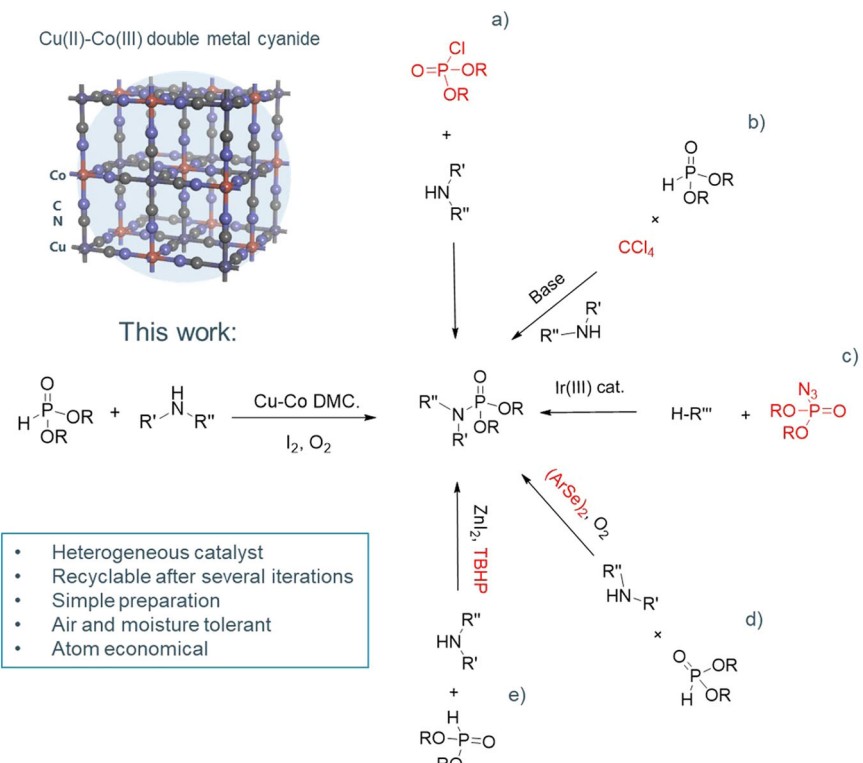

**Fig. 1 Strategies for the synthesis of phosphoramidates.** Previously reported: **a** direct phosphoryl chloride nucleophilic substitution[8,9], **b** Atherton–Todd reaction with the in situ formation of phosphoryl chloride[10], **c** phosphoryl azide C-H amidation[11], **d** diaryldiselenide catalyzed cross-dehydrogenative nucleophilic functionalization[13], and **e** $ZnI_2$ catalytic oxidative coupling[14]. This work: Cu-Co double metal catalyst for the aerobic oxidative coupling of phosphites and amines.

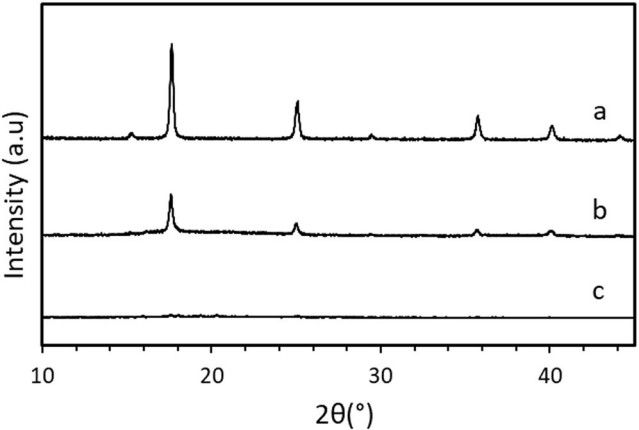

**Fig. 2 X-ray diffractograms of Cu-Co DMC samples. a** pristine Cu-Co DMC (before reaction), **b** washed Cu-Co DMC (recovered after reaction), **c** spent Cu-Co DMC (after reaction).

aerobic oxidation of oximes to ketones, where among a number of mixed metal hexacyanocobaltates, based on Cu, Fe, or Ni, stood out as active catalysts for this transformation, paving the way for their use in aerobic oxidation reactions[34].

Here, we report a heterogeneous, $Cu_3[Co(CN)_6]_2$-based (Cu-Co-DMC) catalytic system for the coupling of phosphites with amines in green synthesis of phosphoramidates, using oxygen and iodine as additives. Remarkably, the Cu-Co-DMC system resulted to be highly active for this reaction and is an excellent alternative for the formation of P(O)-N bonds.

## Results and discussions

**Catalyst characterization**. The characterization of the Cu-Co DMC was in agreement with previous reports[35]. The relative intensities and peak positions of the PXRD data corresponded to the face-centered cubic structure pattern, typical for $M_3[Co(CN_6)]_2$-type DMCs (Fig. 2a)[36]. X-ray diffraction measurements collected after placing the sample in a glass capillary and Pawley fitting (Fig. S1 SI and Table S1 SI) allowed the refinement of the lattice parameters of the Cu-Co DMC, which was found to crystallize in the cubic space group *Fm-3m*. Nitrogen physisorption studies revealed a type 1 isotherm (Fig. S2 SI and Table S2 SI), characteristic of microporous materials, with a Brunauer–Emmett–Teller specific surface area of 660 m$^2$/g, a pore volume of 0.22 cm$^3$/g and an average pore diameter of about 6 Å. ICP-OES analyses show a Cu:Co ratio of 1.7 (Table S3 SI), higher than the stoichiometric 1.5, indicating an excess of Cu ions in the crystalline structure. This slight excess is expected, considering the 10:1 Cu:Co molar ratio employed during the synthesis of the Cu-Co DMC[37]. The leaching of the Cu metal ions after the reaction was also determined and will be discussed in the coming sections (Table S4 SI). The FTIR spectrum of the catalyst showed a blue shift in the position of the CN stretching band compared to the potassium hexacyanocobaltate precursor salt, indicating the formation of the CN-Cu bond (Fig. 3b)[38]. The acid properties of the Cu-Co DMC were studied with pyridine adsorption followed by FTIR spectroscopy. Bands at 1450, 1490, and 1610 cm$^{-1}$ were attributed to the adsorbed pyridine on Lewis acid sites (Figs. S3, S4 SI). No band was observed around 1540 cm$^{-1}$, indicating the absence of Brønsted acid sites on the surface of the catalyst[39]. Thermogravimetric analyses (Fig. S5 SI) revealed that the material is stable up to 300 °C, at which temperature the decomposition of the cyanide ligands begins.

**Reaction optimization**. As a model reaction, the coupling of 2-phenylethylamine and dibutyl phosphite was initially studied. The results of the optimization experiments are presented in Table 1. First, a control experiment was performed in the absence of a catalyst (blank reaction, **entry 1**, Table 1). As expected, the desired phosporamidate was not obtained in measurable amounts. Next, in an initial catalyst screening, a variety of homogeneous and heterogeneous Cu-based catalysts were tested (see Supplementary Methods of the SI for the preparation of the heterogeneous catalysts). In the presence of CuI, the reaction proceeded smoothly in acetonitrile (ACN) (entry 1, Table S5 SI), achieving full conversion in 3 h, which is in agreement with the results obtained by Fraser et al.,[17] albeit at a lower reaction temperature. However, for all the other initially screened catalysts, only poor to mediocre conversions were obtained (entries 2–6, Table S5 SI) (entry 2, Table 1). This suggests that Cu atoms alone are not enough to effectively catalyze the formation of phosphoramidates. In this sense, a more recent report showed that the presence of catalytic amounts of iodine in the reaction was highly beneficial for the overall yield[14]. In their study with dialkyl phosphine oxide as the substrate, an organic oxidant was used to oxidize iodide anions present in solution to generate I$_2$, which reacted to form an iodoamine intermediate. The aforementioned intermediate was then able to iodinate the dialkyl phosphine oxide; after a nucleophilic substitution of the P-I moiety with the available free amine, the desired phosphinic amide was formed (Fig. 4).

In accordance with the latter report, the addition of small amounts of I$_2$ to our reaction resulted in a great increase in yield, from 4 to 86% when Cu(OAc)$_2$ was used as a catalyst (entry 7, Table S5 SI). To confirm that this increase in yield was not exclusively due to the action of the additive (I$_2$), but to the metal-I$_2$ cooperation, a control experiment was done in the absence of the metal catalyst (entry 3, Table 1), resulting in a nearly stoichiometric yield (15%) related to the amount of I$_2$ added (20 mol%), which indicates that a metal catalyst is needed to reach higher conversions. Furthermore, these results explain the difference in catalytic activity between the CuI and Cu(OAc)$_2$ salts; in the case of Cu(OAc)$_2$, the absence of an I species greatly prevents the reaction from proceeding smoothly.

The effect of the O$_2$ pressure was then studied. It was observed that by simply bubbling O$_2$ into the reaction medium using a needle and a balloon, instead of using an open vial with air as the oxidant, the reaction time needed to achieve full phosphite conversion was greatly reduced, from 3 to 0.5 h, without affecting the selectivity to the desired phosphoramidates (entry 8, Table S5 SI); none of the potential oxidation-derived side products were detected by increasing the available O$_2$ in the reaction[17].

In an attempt to design an efficient, heterogeneous system, a series of solid catalysts were screened next (entries 9–12, Table S5 SI). Cu-Co DMC emerged as the best of the screened heterogeneous catalysts. However, preliminary experiments using the Cu-Co DMC yielded the phosphoramidate only in moderate amounts (entry 4, Table 1). Remarkably, the use of less coordinating solvents proved to be favorable for the performance of the catalyst, as can be seen in entries 5–9, Table 1, suggesting that acetonitrile blocks to some extent the available active sites of the DMC, diminishing its overall catalytic activity. In all the tested non-polar solvents, the DMC showed excellent results; however, with dichloromethane (DCM) the highest phosphoramidate yield was obtained (98%). Based on these results and the solubility of most amines and phosphites, DCM was selected as a solvent in further optimization experiments.

The effect of catalyst loading was then studied. Reducing the amount of Cu-Co DMC catalyst used in the reaction from 3 to 1 mol% resulted in a decrease in phosphoramidate yield from 98

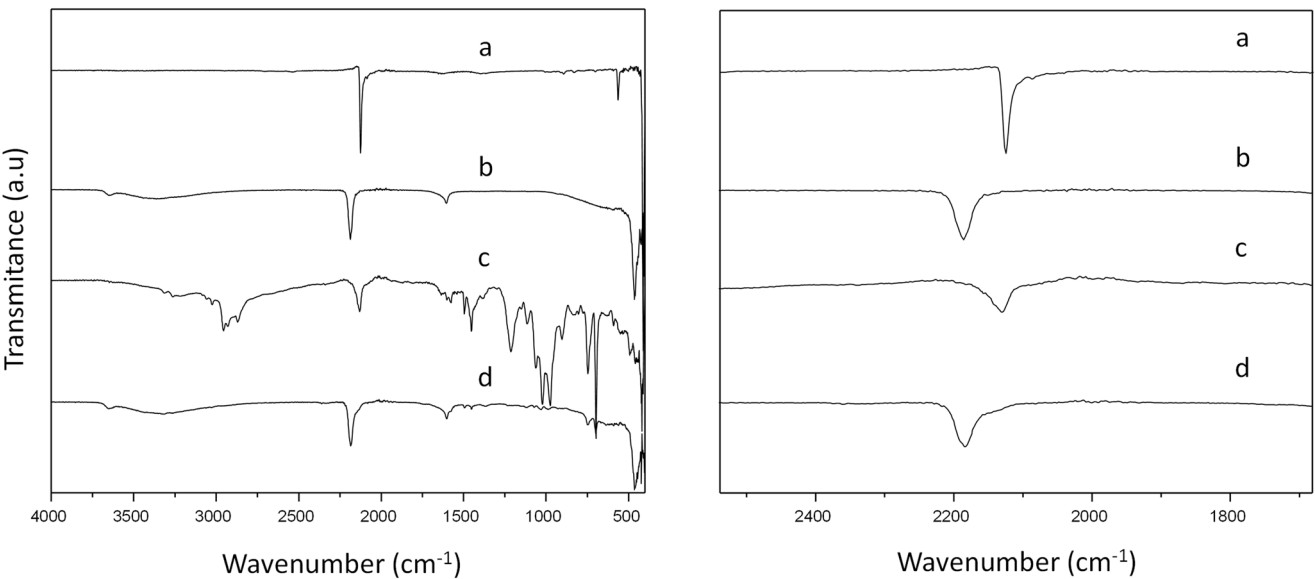

**Fig. 3 FTIR spectra of selected Cu-Co DMC samples. a** $K_3CO(CN)_6$, **b** Pristine Cu-Co DMC, **c** Spent Cu-Co DMC, **d** recovered Cu-Co DMC) with zoom on the C≡N stretching bond region.

---

**Table 1 Reaction optimization[a,b].**

| Entry | Catalyst (mol%) | Solvent | Phosphite/Amine/I₂ (equivalents) | Source of O₂ | Yield |
|---|---|---|---|---|---|
| 1 | No Cat. | ACN | 1/2/0 | Air | >1%[c] |
| 2 | Cu-Co-DMC (3%) | ACN | 1/2/0 | Air | 6%[c] |
| 3 | No Cat | ACN | 1/2/0.20 | Air | 15%[c] |
| 4 | Cu-Co-DMC (3%) | ACN | 1/2/0.2 | O₂ balloon | 49% |
| 5 | Cu-Co-DMC (3%) | THF | 1/2/0.2 | O₂ balloon | 90% |
| 6 | Cu-Co-DMC (3%) | Dioxane | 1/2/0.2 | O₂ balloon | 89% |
| 7 | Cu-Co-DMC (3%) | DCM | 1/2/0.2 | O₂ balloon | 98% |
| 8 | Cu-Co-DMC (3%) | Toluene | 1/2/0.2 | O₂ balloon | 84% |
| 9 | Cu-Co-DMC (3%) | 2-MeTHF | 1/2/0.2 | O₂ balloon | 90% |
| 10 | Cu-Co-DMC (1%) | DCM | 1/2/0.2 | O₂ balloon | 72% |
| 11 | Cu-Co-DMC (6%) | DCM | 1/2/0.2 | O₂ balloon | 99% |
| 12 | Cu-Co-DMC (3%) | DCM | 1/2/0.15 | O₂ balloon | 99%[d] |
| 13 | Cu-Co-DMC (3%) | DCM | 1/2/0.1 | O₂ balloon | 87% |
| 14 | Cu-Co-DMC (3%) | DCM | 1/2/0.05 | O₂ balloon | 53% |
| 15 | Cu-Co-DMC (3%) | DCM | 1/1/0.15 | O₂ balloon | 42% |
| 16 | Cu-Co-DMC (3%) | DCM | 1/3/0.15 | O₂ balloon | 90% |
| 17 | Cu-Co-DMC (3%) | DCM | 1/2/0.15 | Air | 96%[c] |
| 18 | Cu-Co-DMC 1.5 (3%) | DCM | 1/2/0.15 | O₂ balloon | 90% |
| 19 | No Cat | DCM | 1/2/0.15 | O₂ balloon | 12%[c] |

[a]All reactions were performed using dibutyl phosphite as a limiting reagent at a scale of 2 mmol, amine, catalyst, solvent (4 ml), iodine, and a source of oxygen at room temperature for 0.5 h.
[b]Yields of dibutyl phenylethyl phosphoramidates were determined by $^1$H NMR spectroscopy using 1,3,5-trimethoxybenzene as internal standard.
[c]Reaction time 3 h.
[d]Turnover frequency (TOF, mol of phosphoramidate produced per mol of Cu per hour) was 266.7 h$^{-1}$, based on initial reaction rates (0.5 min).

---

to 72% (entry 10, Table 1), whereas an increase in catalyst loading to 6 mol%, did not result in a decrease in reaction time needed to achieve full phosphite conversion (entry 11, Table 1). The effect of I₂ concentration was also studied. The amount of I₂ used in the reaction could be reduced to 15 mol% without negatively affecting the phosphoramidate yield (99%, entry 12, Table 1), however, employing lower amounts (10 mol% or lower), resulted in a decrease of the phosphoramidate yield (entries 13 and 14, Table 1). Finally, the phosphite:amine molar ratio was set to 1:2, since adding stoichiometric amounts of each (entry 15, Table 1), or a larger excess of amine (entry 16, Table 1) appeared to be detrimental toward the phosphoramidate yield. Conditions in entry 12 of Table 1 were set as the newly optimized conditions for the reaction.

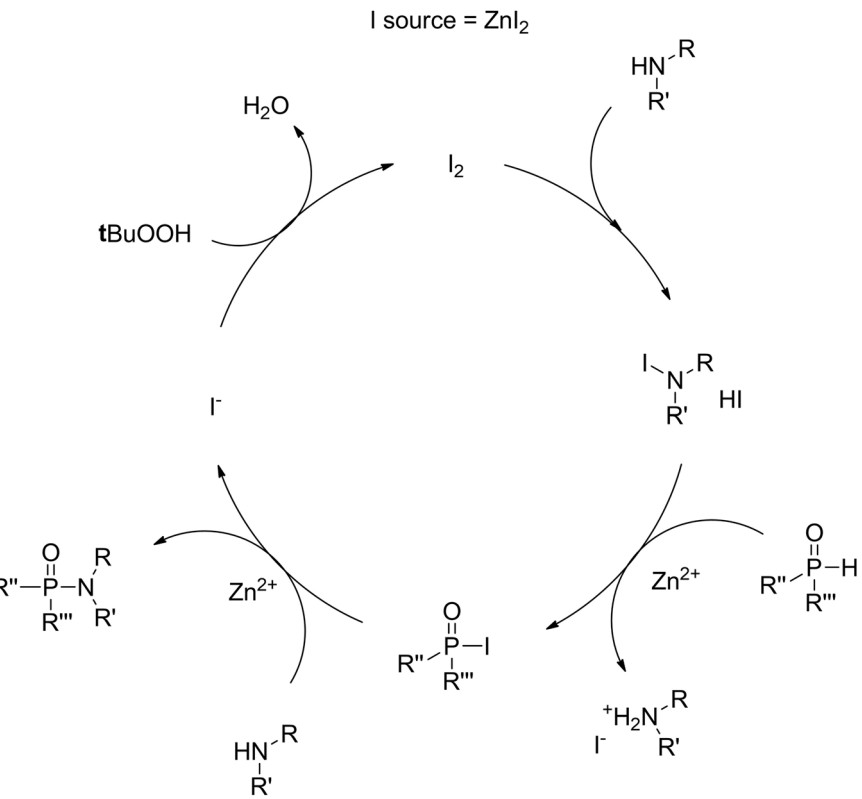

**Fig. 4 I$_2$ catalytic redox cycle for the formation of phosphoramidates using an organic oxidant[14].** Iodine is formed by the addition of stoichiometric amounts of *tert*-Butyl hydroperoxide using the available iodide in solution.

Under optimized reaction conditions, a different heterogeneous catalyst (Cu-BTC) was also able to achieve excellent phosphoramidate yields (entry 9, Table S5 SI), even comparable to those obtained with the Cu-Co DMC. However, ICP measurements of the final reaction crude solution showed high values of Cu leaching, indicating poor stability of the MOF under these conditions (Table S4 SI). All other tested heterogeneous catalysts still showed poor activity for this transformation (entries 10–12, Table S5 SI). The use of an O$_2$-filled balloon proved beneficial for shortening the reaction times needed under the found optimum conditions. As shown in entry 17 **of** Table 1, when removing the O$_2$-filled balloon, the reaction took 6 times longer to be completed. Noticeably, small amounts of side products were also formed when air (instead of O$_2$) was used as an oxidant. It is also worthwhile to mention the slight differences in phosphoramidate yield depending on the synthesis conditions of the Cu-Co DMC catalyst. For instance, when the DMC was prepared using a stoichiometric Cu/Co ratio (Cu/Co = 1.5) of the precursor metal salts and without the use of any organic additives (*tert*-butanol) (Cu-Co DMC 1.5, entry 18, Table 1), a slightly less active catalyst was obtained, thus showing the beneficial effects of using an excess of one of the metallic salts and of *tert*-butanol as an additive during the synthesis of the Cu-Co DMC catalyst. A final control experiment under the optimized conditions (entry 19, Table 1) showed that only stoichiometric amounts of product were obtained with respect to the amount of I$_2$ added when no catalyst was present in the reaction.

To prove the heterogeneous nature of the Cu-Co-DMC, ICP-OES measurements of the final reaction crude solution were performed. Only 1.6% of the total Cu content leached into the solution from the catalyst (Table S4 SI). A hot filtration test was conducted, also confirming that there is no substantial leaching of catalytically active species from the DMC, and thus, its stability and heterogeneity (Fig. S6 SI). To further confirm the heterogeneous nature and reusability of the catalysis, the coupling reaction was carried out under the optimal conditions described above, and upon completion of the first run, the Cu-Co DMC catalyst was recovered, washed with methanol and water:*tert*-butanol solution, dried at 60 °C overnight and reused in consecutive experiments. The catalyst showed no significant loss of activity, even after 5 iterations (Fig. S7 SI).

Interestingly, PXRD measurements revealed a possible change in the crystalline structure of the Cu-Co DMC catalyst after the reaction. Diffraction patterns of the spent Cu-Co DMC (after the reaction, but before the washing step) showed an almost complete loss of the cubic crystalline structure of the sample; however, this phase could be at least partially recovered after the catalyst is submitted to the washing process (Fig. 2b, c). Similar changes in Cu-Co-based DMCs during reaction have been observed in the synthesis of propargylamines through A$^3$ coupling, where a comparable loss of crystallinity or phase change was reported[40]. Furthermore, it is important to note that the catalyst in that case also retained its catalytic activity for at least three runs after the loss of crystallinity/phase change. Therefore, additional recycling experiments using the catalyst directly after the reaction (no washing step) were also carried out. These experiments showed that both the spent Cu-Co DMC (after reaction) and the washed Cu-Co DMC (recovered after reaction) were equally active even after five reaction iterations (Figs. S7, S8 SI), which suggests that the crystallinity is not the key factor for the catalytic activity of Cu-Co DMC. In this regard, examples of active yet amorphous DMCs have also been reported in the literature; specifically, for amorphous Zn-based DMCs, which have been extensively used for the ring-opening polymerization of propylene oxide[20,41,42].

Similarly, FTIR measurements of the aforementioned samples displayed intriguing differences between them. Specifically, the characteristic stretching band of the C≡N bond in the Cu-Co DMC catalyst shifted towards lower frequencies after the

completion of the reaction, as depicted in Fig. 3c, which hints at changes in the electronic environment of the C≡N, and hence, of its surrounding metals. After the subsequent washing step (first with methanol and then with water:tert-butanol 50:50 solution), the distinctive signal reverted to its initial position, as seen in Fig. 3d. The similarity in the energy of the C≡N bond following the reaction to that observed in the precursor hexacyanocobaltate salt (Fig. 3a), led us to speculate that the nature of the bridging between $M^1$-CN-$M^2$ in the DMC may have changed after the reaction, thereby resulting in the loss of crystallinity. Therefore, the local structure of the metal centers was further studied in the last section of this work.

**Reaction scope**. The scope of the reaction was extended to the synthesis of different phosphoramidates, and the results are presented in Table 2. Under these oxidative conditions, one might expect the decomposition of amines (formation of the corresponding aldehyde, carboxylic acid, or dialkylamine), decreasing the overall production of the desired product. Remarkably, this was not the case for most of the studied substrates. The major recurring side product observed in most of the reactions was identified as the corresponding phosphate (see [1]H and [31]P NMR spectra of the SI). Efforts to minimize the formation of this impurity employing anhydrous solvents were unsuccessful, proving that the formation of this product was not caused by the presence of moisture in the reaction, but by oxidation of the starting phosphite.

Dibutyl phosphite was successfully coupled with propylamine and benzylamine in very good yields (entries 2 and 5, Table 2), and only small amounts of oxidation side products were found in the final crude. Experiments carried out with piperidine, a cyclic, secondary amine, only formed the phosphoramidate in moderate yields under these conditions, most likely because of steric hindrance, as shown in entry 4 of Table 2. Coupling of phosphites with the studied primary amines worked well in this method, as very good yields are obtained with the exception of allylamine (entry 3, Table 2). In fact, even bulkier primary amines, such as isopropylamine, could also be incorporated in the corresponding phosphoramidate in good yields (entry 6, Table 2), indicating, to some extent, tolerance to the size of the coupling partner. This method could potentially also be expanded to amides, such as benzylamide as N-source; this reaction yields dimethyl benzoyl-phosphoramidate in high yields only after 0.5 h of reaction time (entry 9). Reducing the length of the phosphite alkyl chain had an adverse effect on the reaction in the case of the MeO— groups, as shown in entries 1 vs. 7 and 8 of Table 2. This seems counterintuitive at first glance, since shortening the chains would make the substrate even more accessible to the amine, however, this increase in accessibility would also make it more susceptible to other attacks as seen by the presence of relevant amounts of side product in the final reaction crude, therefore, decreasing the selectivity to the desired phosphoramidate. The reaction also proceeded smoothly when aromatic amines were used as coupling partners (entries 10–12, Table 2). As expected, the presence of substituents on the aromatic ring changed the overall performance of the reaction, where no substituents were preferred in order to achieve higher yields. Noticeably, strong *para* electron-donating groups on the aromatic ring seem to be less detrimental to the reaction than weakly electron-donating groups (entry 11 vs 12, Table 2). The formation of phosphoramidates from L-alanine and L-aspartic acid esters (entries 13 and 14, Table 2) showed the tolerance of these groups to the reaction conditions, as no deprotection of the carboxylic acid groups was seen during the transformation. This suggests that this methodology could be used to prepare other amino acid derivates and potentially

nucleoside conjugates. An industrially relevant type of phosphorous-based flame retardant, a DOPO-derivate, was also synthetized in respectable yield using this methodology (entry 15, Table 2). To further prove the industrial potential of this method, we were able to successfully scale up entry 1 of Table 2 to a 20 g batch without negatively affecting the yield; after quenching the reaction with an aqueous saturated solution of sodium thiosulphate, washing with diluted HCl and sodium bicarbonate, drying with anhydrous sodium sulfate and in vacuo removal of volatiles, the final product was obtained in high purity and an isolated yield of 93%.

**Mechanistic study**. Considering the results obtained during the optimization, it was hypothesized that the Cu-Co DMC catalyst was able to promote the reaction by oxidizing $I^-$ anions to molecular $I_2$ in the presence of oxygen. To prove this, the iodide oxidation in the presence of the DMC catalysts was studied and the results are presented in Table 3. To this end, 40 mg of Cu-Co DMC were added to a 0.5 M methanolic solution of NaI. Methanol was chosen as a solvent instead of DCM as it is able to easily dissolve the NaI salt, which is insoluble in most organic solvents. After stirring for 2.5 h in the open air, only small amounts of $I_2$ were detected (5% yield, entry 1). However, after the addition of a small amount of HCl a fourfold increase in the $I_2$ yield was obtained (20%, entry 2, Table 3) (Figs. S9, S10 SI). Oxidation of $I^-$ in the absence of the Cu-Co DMC under acidic conditions yielded only small amounts of $I_2$ (entry 3, Table 3), showing that the catalyst is indeed crucial to increase the reaction rate.

Experiments with larger amounts of HCl, showed higher $I_2$ yields for $H^+$ concentrations up to 100 mM. After that point, further acidification of the solution did not result in any notable increase in conversion (Fig. S11 SI). The stability of the catalyst under the studied acid conditions was evaluated by XRD. The different diffraction patterns of the Cu-Co DMC sample after stirring in methanolic solutions at different concentrations of HCl (75, 150, and 225 mM) for a period of 24 h revealed the stability of the said catalyst, as they showed little to no differences between them (Fig. S12 SI). Additionally, ICP-OES analyses were conducted, which revealed that the Cu content of the final crude solution was below the detection threshold (5 ppb), further proving the stability of the catalyst.

These results confirm our hypothesis that the Cu-Co DMC can catalytically oxidize iodide, as proposed in the reaction mechanism (Fig. 5). Here, the active Cu sites regenerate the consumed $I_2$ in the reaction and reduce the present molecular oxygen. This oxidation-reduction cycle is directly coupled to the phosphoramidate formation. First, a iodoamine intermediate is generated by the reaction of $I_2$ with the free available amine. Acting as an iodination agent, the iodoamine then reacts with one of the phosphite's tautomeric forms, giving the highly reactive dialkyl iodophoposphate. The formation of the desired phosphoramidate is finally achieved by the nucleophilic attack of the amine to this active phosphinic iodide.

**Local structure of the metal centers**. The local environment around the Co and Cu ions in the studied Cu-Co DMC was thoroughly characterized by Co K-edge and Cu K-edge XANES and EXAFS measurements. The study was focused on three different stages of the catalyst: pristine Cu-Co DMC (before reaction), spent Cu-Co DMC (after reaction), and washed Cu-Co DMC (recovered after reaction), with special consideration to the intriguing changes in the structure of the catalyst revealed by PXRD studies. Results of the analyses for the pristine Cu-Co DMC (before reaction) are presented in Fig. 6. For Co atoms, the edge

**Table 2 Reaction scope of the phosphoramidate formation with Cu-Co DMC catalysts[a].**

| Entry | Phosphite | Amine | Product | Yield[b] |
|---|---|---|---|---|
| 1 | | | | 99% |
| 2 | | | | 81% |
| 3 | | | | 21% |
| 4 | | | | 26% |
| 5 | | | | 89% |
| 6 | | | | 82% |
| 7 | | | | 94% |
| 8 | | | | 71% |
| 9 | | | | 61% |

**Table 2 (continued)**

Cu-Co DMC (3 mol %)
$I_2$ (15 mol %)

$O_2$ (1 atm)
DCM (0.5 M)
RT, 0.5 h

| Entry | Phosphite | Amine | Product | Yield[b] |
|-------|-----------|-------|---------|----------|
| 10 | | | | 75% |
| 11 | | | | 68% |
| 12 | | | | 59% |
| 13 | | | | 70% |
| 14 | | | | 64% |
| 15 | | | | 58% |

[a]Scope of the Cu-Co DMC catalyzed aerobic oxidative coupling. Reaction conditions: Phosphite (0.20 mmol) and amine (0.40 mmol).
[b]Yields were determined based on characteristic peaks of phosphoramidates by [1]H NMR spectroscopy using 1,3,5-trimethoxybenzene as an internal standard.

position in the XANES region indicated that Co is present as $Co^{3+}$ species throughout the catalyst lattice (Fig. 6a and Fig. S13a SI). Being sensitive to the exact type of ligands, XANES spectra also showed a linear Co-C-N-Cu (and not, for example, Co-N-C-Cu) disposition of the $CN^-$ bridges (Fig. S14 SI). Co K-edge EXAFS data (Fig. 6c) confirmed an octahedral surrounding for Co, with clearly observed first (C), second (N), and third (Cu) shells, in agreement with the cubic structure identified by PXRD. High intensity of the second shell signal was noticeable, mainly due to multiple scattering contributions by the linear arrangement of the cyanide ligand. Thus, data showed a full octahedral cyanide-bridged coordination in a cubic model, leaving no free vacancies on the coordination sphere, and therefore, no available catalytic sites on the Co atoms. The inherent strong bonding nature of the $CN^-$ ligand by σ-donation and π-backbonding, and the great stability of the octahedral hexacyanocobaltate complex are also in

**Table 3 Iodide oxidation mediated by Cu-Co DMC[a].**

NaI $\xrightarrow[\substack{MeOH \\ Air,\ rt,\ 2.5\ h}]{Cu\text{-}Co\ DMC,\ HCl}}$ I$_2$

| Entry | Cu-Co DMC (mg) | H$^+$ (mM) | Yield[b] |
|-------|----------------|-----------|----------|
| 1 | 40 | 0 | 5% |
| 2 | 40 | 75 | 20% |
| 3 | 0 | 75 | 8% |

[a]All reactions were performed using a scale of 2 mmol of NaI, catalyst, 37% HCl, and methanol as solvent (4 ml) at room temperature, exposed to air, for 2.5 h.
[b]Yields were determined by measuring I$_3^-$ via UV-Vis measurements ($\lambda = 360$ nm) (Supplementary Note 1, Figs. S9, and S10 SI).

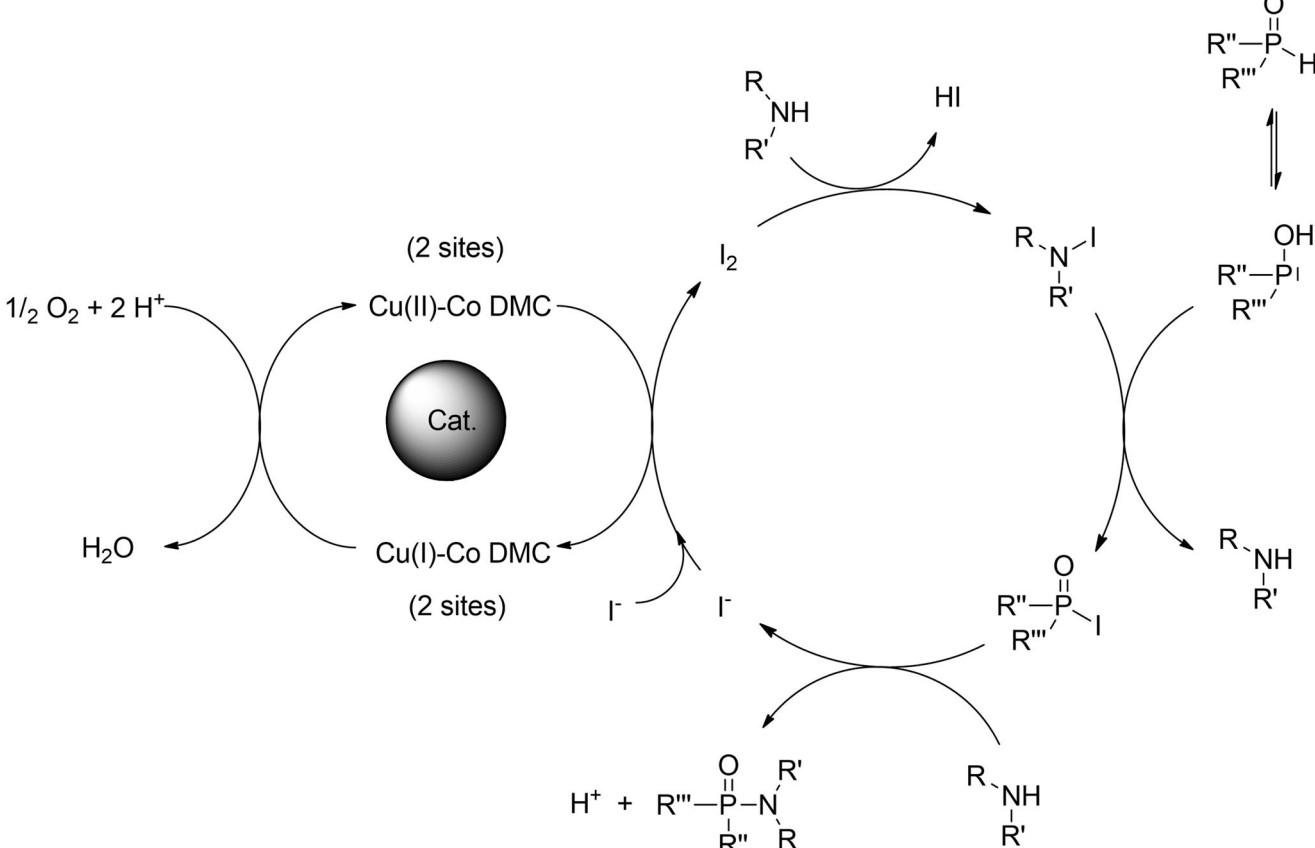

**Fig. 5 Proposed reaction mechanism for the Cu-Co DMC catalyzed aerobic oxidative coupling of phosphites and amines.** The Cu-Co DMC catalyst completes the iodine catalytic cycle consuming oxygen and forming water as the only by-product.

line with this hypothesis, where the primary role of the [Co(CN)$_6$]$^{3-}$ anions is to act as building blocks inside the Cu-Co DMC catalyst[31].

Catalysis would then most likely only occur on the available open coordination sites associated with Cu. Mullica et al.[36] proposed a model for the cubic (space group *Fm-3m*) type structure for DMCs, where vacancies in the octahedral lattice of one of the metals are necessary to maintain charge balance. In their study, the different charges of the [Co(CN)$_6$]$^{3-}$ anions and the Zn$^{2+}$ cations created defects in the crystalline structure, leaving open sites on the coordination sphere of the Zn atoms. The Cu *K*-edge data revealed an analogous behavior in the case of our Cu-Co DMC. Although the three-shell fitting shown in Fig. 6d confirms the same cyanide-bridged model applied for Co, a more careful

analysis of the first shell (Fig. S13 SI and Table S6 SI) exposed a coordination number below 6, lower than expected in a vacancies/defect-free structure. In this regard, an average coordination number for Cu atoms of 4 ± 1 (Table S6 SI), together with XANES data confirming an octahedral-like geometry (Fig. 6b), unequivocally points to the existence of hexacyanocobaltate vacancies. The available free sites in the coordination sphere of the Cu are then believed to be responsible for the catalytic activity of the Cu-Co-DMC. It is worth mentioning that DMCs are known to be pure Lewis acid catalysts (as seen in the FTIR measurements after adsorption of pyridine); therefore, in this case, the Lewis acidity can be attributed to such vacancies.

Measurements of the spent Cu-Co DMC sample (after reaction) showed almost no change in both local electronic and

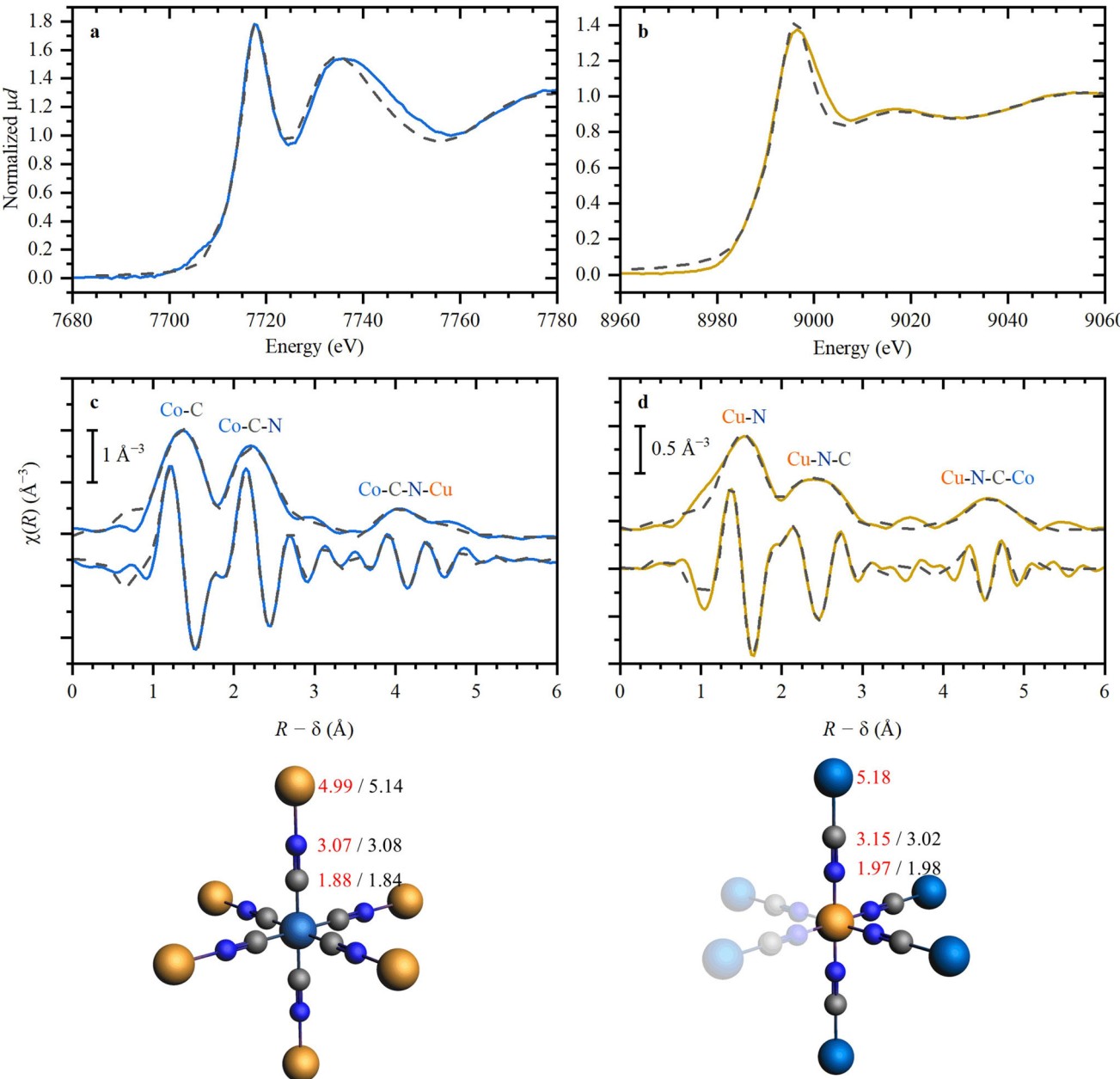

**Fig. 6 XANES and EXAFS data for Co and Cu K-edges.** XANES (**a**, **b**), phase-uncorrected EXAFS (**c**, **d**); Co (**a**, **c**), Cu (**b**, **d**). Experimental data is shown by colored solid lines, and the dashed black lines correspond to the best fit. In the proposed structural models, interatomic distances obtained from EXAFS and XANES are shown in red and black, respectively. Atomic color code: Co – blue, Cu – dark yellow, C – gray, N – dark blue. Numbers in the atomic structures correspond to the distance from the absorbing atom (Co or Cu) to the corresponding atom obtained by EXAFS (in red) and XANES (in black) fitting.

atomic structure of Co centers, except for some loss of Co-Cu coordination (third shell) (Fig. 7a, c), suggesting that fewer Cu atoms are coordinated to the hexacyanocobaltate units than in the pristine Cu-Co DMC. Taking into account the ICP measurements indicating no Cu leaching, these results then suggest a loss of long-range order inside the catalyst structure. This structural change is also suggested by the FTIR data of the studied sample, which shows a red shift in the position of the C≡N band (Fig. 3c). Furthermore, as shown by PXRD, the crystalline cubic structure of the pristine catalyst was lost, and a different (amorphous) phase was formed. However, XAS data revealed that the local structure of Co was similar to the initial state, with $Co^{3+}$ ions remaining in the octahedral coordination connected to the $CN^-$ ligands. Cu K-edge XANES (Fig. 7b), was characterized by an edge shift towards slightly lower energy and a decrease of the first

maximum, hinting at the presence of a softer ligand. The almost complete loss of Cu-Co contribution in the third shell and the significant changes in the second shell are evident from EXAFS (Fig. 7d), and also suggest a different type of ligands for Cu after reaction. The identification of the ligand type is quite ambiguous in this case; therefore the shown red atoms in the Cu surroundings in Fig. 7 (bottom right) may correspond either to O or to N. Various examples of XANES fittings with various ligands are shown in Fig. S15 of the Supporting Information for comparison. N atoms may originate from a combination of the N end of the $CN^-$ ligand in the DMC structure and the N atoms from phenylethylamine present in the reaction media, whereas O atoms could come from water molecules, dibutyl phosphite or dibutylphosphoramidate. Moreover, it is worth considering that due to the oxidizing conditions of the reaction, the formation of a

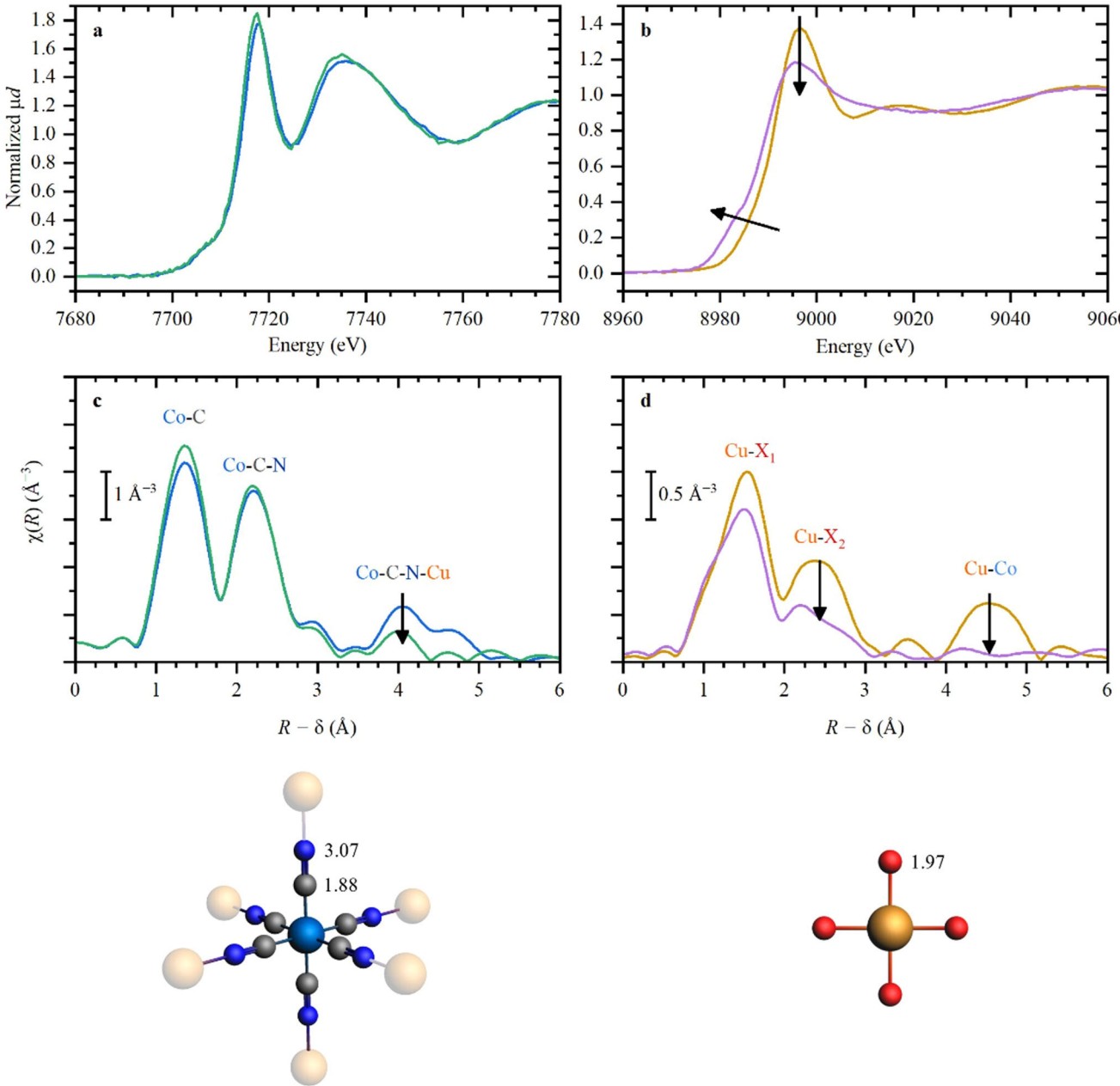

**Fig. 7 XANES and EXAFS data for Co and Cu K-edges before and after reaction.** Experimental XANES (**a**, **b**) and phase-uncorrected EXAFS (**c**, **d**) data for Co (**a**, **c**) and Cu (**b**, **d**) K-edges before (blue and orange) and after (green and purple) reaction. The arrows indicate the observed spectral changes. X atoms highlighted in red correspond to unidentified light neighbor: C, N, or O.

CuO phase (Cu coordinated to 4O atoms) could not be excluded. However, the comparison of the catalytic activity of CuO (entry 10, Table S5 SI), and this Cu phase formed after the reaction showed a marked superior activity of the latter phase, which discredits this hypothesis.

Finally, XAS measurements for the washed Cu-Co DMC (recovered after reaction) sample were in agreement with its respective PXRD diffractogram, presented in Fig. 2, and with its FTIR spectrum (Fig. 3d). A partial recovery of crystallinity and a regression to the original cubic phase could be seen after washings with methanol and water:*tert*-butanol, reflected in the growth of the first maximum in XANES (Fig. 8). However, the resulting spectrum after washing did not share isosbestic points with the initial and spent states of the catalyst, indicating the presence of an additional phase, which was identified as tetrahedral Cu.

Indeed, the addition of $Cu_2O$ as a reference for tetrahedral coordination results in a perfect reconstruction of the spectrum of the washed catalyst by a linear combination of three phases: square planar, as in the spent catalyst (13%), octahedral, as in the initial catalyst (59%), and tetrahedral (27%). This tetrahedral geometry of the $M^1$ metal in DMCs is not necessarily exceptional, and has been previously reported in the literature[32,33]. In the case of the Zn-Co DMC, the Zn atom could be found tetrahedrally coordinated to the N end of the CN bond if the ambient humidity is low enough or if other coordinating organic molecules are present in the medium[43,44]. This would result in a material with a rhombohedral or a monoclinic phase and a distinct crystalline PXRD pattern, different from that of the commonly reported cubic phase. In our case however, no peaks attributed to either of the aforementioned crystalline phases (space groups *R-3c* and

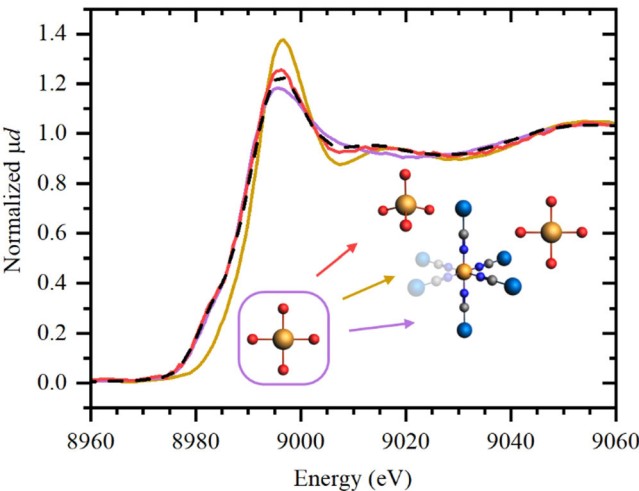

**Fig. 8 Experimental Cu K-edge XANES spectra for initial, spent and washed catalyst.** Experimental Cu K-edge XANES spectra for initial (dark yellow), spent (purple) and washed (red) catalyst. The dashed line corresponds to the linear combination fit of the spectrum of the washed state using three experimental spectra (initial, spent, and tetrahedral $Cu_2O$).

$P2_1/m$, respectively) are observed in the PXRD pattern of the washed Cu-Co DMC (recovered after the reaction), suggesting that the tetrahedrally coordinate Cu atoms could be either part of a non-crystalline phase, or that the contribution of the amorphous phase is superimposed over the peaks of the other phase. Notably, the Co atoms remained unchanged in their initial octahedral geometry.

## Conclusions

We reported the development of a method for the efficient synthesis of phosphoramidates using a heterogeneous and recyclable Cu-Co DMC catalyst. Catalytic amounts of molecular iodine were used to produce an in situ iodophosphate intermediate, which would later lead to the formation of the P-N bond; the catalytic cycle is closed by aerobic oxidation involving the metallic center of the DMC. Advanced characterization by PXRD, FTIR and XAS, confirmed a change in the coordination geometry of the metal centers due to the interaction of the Cu atoms with the reagents and the redox nature of the reaction, and the formation of two different phases during its reuse. However, the observed changes did not have an impact on the remarkable catalytic activity of the Cu-Co DMC. All in all, the system proposed in this work provides a sustainable and reliable methodology for obtaining high-value molecules and contributes to the expansion of the reaction scope of DMCs as oxidative coupling catalysts.

## Methods

**Synthesis of Cu-Co DMC**. The Cu-Co DMC was synthesized by modifying previously reported procedures[44]. About 15 ml of a 0.1 M aqueous solution of $K_3[Co(CN)_6]$ were added dropwise, under continuous stirring, to 150 ml of a 0.1 M aqueous solution of $CuCl_2 \cdot xH_2O$. After both solutions were mixed, 37.5 ml of *tert*-butanol were added, and the mixture was stirred for 3 h at room temperature. The formed precipitate was recovered by centrifugation (10,000 rpm for 10 min at room temperature), washed three times with 50 ml of a 1:1 mixture of water:*tert*-butanol, and dried at 110 °C overnight. The catalyst was used without any further pretreatment or pre-activation before the reaction.

**Characterization**. The Cu-Co DMC catalyst was characterized using Powder X-ray diffraction (PXRD), inductively coupled plasma optical emission spectroscopy (ICP-OES), $N_2$ physisorption, Fourier-transform infrared spectroscopy (FTIR), thermogravimetric analysis (TGA), X-ray absorption near edge structure (XANES),

and extended X-ray absorption fine structure (EXAFS) spectroscopy. PXRD measurements were performed on a Malvern PANalytical Empyrean diffractometer using a PIXcel3D solid-state detector and a Cu anode (Cu $K_{\alpha1}$ = 1.5406 Å; Cu $K_{\alpha2}$ = 1.5444 Å) operating at 40 mA and 45 kV with a focusing X-ray mirror module. PXRD patterns were collected in transmission mode over a 2θ range of 1.2° < 2θ < 45° with a step of 0.0131° and a counting time of 100 s. In order to suppress the fluorescence contribution to the background in Co-containing samples, the pulse height distribution (PHD) settings of the detector were adjusted to a range of 45–75%. An additional pattern was collected between 10° and 100° 2θ, placing the sample in a glass capillary. The obtained diffraction pattern was then analyzed using the JANA-2006 software package[45], and the lattice parameters were refined by Pawley-type fitting[46]. The metal ratio of the DMC was determined with ICP-OES using a Varian 720-ES equipped with a double-pass glass cyclonic spray chamber, a Sea Spray concentric glass nebulizer, and a high solids torch. Samples were digested in a 7:3 (v:v) solution of $HNO_3$-HCl and heated to 200 °C in a microwave oven for 2 h. $N_2$ physisorption isotherms were collected on a Micromeritics 3Flex Surface Analyzer at −196 °C. Before the measurements, the samples were evacuated at 120 °C for 16 h. The specific surface area ($S_{BET}$) was determined using the BET method in a $p/p^0$ range from 0.004 to 0.03. The specific external surface area ($S_{ext}$) and the micropore volume ($V_{micro}$) were obtained using *t*-plot analysis. The median pore size was determined using the HK method. The FTIR spectra of the samples (KBr wafers, ~1 wt.% of sample) were collected in ATR mode on an Agilent Cary 630 FTIR. Pyridine adsorption followed by FTIR spectroscopy was used to determine the acid nature and acid site density of the catalyst using a Nicolet 6700 FTIR spectrometer. For this, a self-supported wafer (~10 mg cm$^{-2}$) was placed in a cell under a vacuum and heated at 250 °C for 1 h. The cell was further cooled down and pyridine (25 mbar) was adsorbed onto the wafer at 50 °C until sample saturation. Subsequently, a reference spectrum of the sample was recorded at room temperature. The weakly coordinated superficial pyridine was removed by evacuation for 30 min before recording the IR spectrum of the strongly coordinated pyridine at 150 °C. The Lewis acid site density was then calculated from the 1450 cm$^{-1}$ absorption band area in the difference spectrum using the integrated molar extinction coefficient from Emeis[47]. TGA measurements were carried out on a TGA Q500 of TA Instruments (10 °C/min heating rate, compressed air atmosphere). XANES/EXAFS characterization at Co and Cu K-edges was done using a laboratory X-ray absorption spectrometer manufactured by Rigaku. All samples were pelletized for measurements in the transmission geometry. For this purpose, the powdered samples were mixed with boron nitride, the masses were calculated by the XAFSmass program, and pressed into rectangular 1.8 × 0.5 mm² pellets. The X-ray tube, equipped with a tungsten cathode and anode, was operated at 14 kV and 40 mA. The measurements were performed in transmission geometry with a Ge (311) Johansson curved crystal as a monochromator, providing energy resolution $\Delta E = 1.9$ eV at Cu K-edge. The beam intensity before the sample was recorded by an ionization chamber filled with Ar under 300 mbar, and an SC-70 X-ray scintillation detector was located after the sample to record the transmitted beam. The data were processed in a standard way in Demeter software[48]. Theoretical Cu K-edge and Co K-edge XANES spectra were calculated using the finite difference method implemented in FDMNES[49,50]. The radius of the shell with atomic clusters inside was chosen as 5.5 Å around the absorbing atom (Cu or Co). Convolution parameters were chosen automatically using PyFitIt to get the best agreement with the experimental spectra[51]. The experimental spectra were fitted to the theoretical ones using a machine learning (ML) approach implementing PyFitIt code. For each hypothetical model of the Cu and Co local structure, 30 deformations of bond lengths were applied to form the training set for the ML algorithm. The best geometry was then obtained by fitting the experimental spectra with ML-predicted ones (Supplementary Notes 2, 3).

### Catalytic reactions

*General methods.* All reagents and starting materials were obtained commercially from Sigma-Aldrich and were used as received without further purification. All reactions were carried out in glass vials or round-bottomed flasks and were monitored by thin-layer chromatography on silica gel 60 F254 coated alumina plates, using potassium permanganate stains for visualization, gas chromatography (GC, Shimadzu 2014 GC equipped with an FID detector and a CP-Sil 5 CB column) and gas chromatography coupled with mass spectrometry (GC-MS, Agilent 6890 gas chromatograph, equipped with an HP-5MS column, coupled to a 5973 MSD mass spectrometer). Additional $^1H$ and $^{31}P$ NMR spectra for product identification and quantification were recorded on a Bruker 400 MHz spectrometer at ambient temperature using [$D_6$] DMSO as solvent (Supplementary Note 4 and Fig. S16 SI).

*General synthesis of phosphoramidates.* To a sealed glass vial with magnetic agitation containing Cu-Co DMC (3 mol%), 2 ml of solvent were added. Subsequently, the corresponding phosphite (2 mmol) was added under continuous agitation. In a separate vial, 2 ml of a 15 mol% $I_2$ solution was prepared, to which the respective amine (4 mmol) was slowly added. The amine-iodine solution was then added dropwise to the phosphite-containing suspension. Finally, an $O_2$-filled balloon was connected to the system, bubbling $O_2$ into the reaction medium through a syringe. The reaction was then left at room temperature for the desired time.

Upon completion of the first run, the Cu-Co DMC catalyst could be separated from the reaction solution by centrifugation, washed first with excess methanol, next with a 1:1 solution of water:tert-butanol, and dried at 60 °C overnight. The recovered catalyst could be then reused in further experiments.

*Oxidation of iodide by Cu-Co DMC.* The oxidative capabilities of Cu-Co DMC were proven by adding the catalyst to an acidic solution of NaI. To an open glass vial with magnetic agitation containing NaI (0.6 mmol), 4 ml of MeOH were added. Next, a desired amount of Cu-Co DMC was added after the dissolution of the salt. Finally, the reaction was left to react for 2.5 h at room temperature after the addition of varying amounts of 37% HCl.

## Data availability

The corresponding data of this study is available within the paper and the supplementary information. Any additional raw data files needed are available from the corresponding author upon reasonable request.

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

## Acknowledgements

This work has been performed in the framework of the C-Planet project which has received funding from the European Union's Horizon 2020 research and innovation program under the Marie Sklodowska-Curie grant agreement No 859885. D.D.V. and A.L.B. thank FWO Vlaanderen for project support (G0F2320N).

## Author contributions

A.F. designed and performed the experiments, analyzed and interpreted the data, and wrote the manuscript; C.M. and D.D.V. envisioned and designed the experiments, supervised the study, and assisted with the writing and editing of the manuscript at all stages. A.L.B. and A.Y.P. performed the XAS measurements and interpreted the results. K.J. was involved in the communication and reporting of the XAS data.

## Competing interests

The authors declare no competing interests.
