## [Peer Review File · Communications Chemistry]

Reviewers' comments:

Reviewer #1 (Remarks to the Author):

The article describes a promising strategy to produce phosphoramidates from amine and phosphite using a sustainable Cu-Co double metal cyanide or Prussian blue analogue (PBA) and oxygen/iodine as additives. The work is innovative and original, expands the scope of DMC catalysis. Considering the novelty of the synthetic route for phosphoramidates, comprehensive studies on the characterization of catalyst, reaction mechanism, and broad substrate scope, this work should be accepted to for publication in Communications Chemistry.

The following comments may help the authors for revising their manuscript:

1. In the preparation of Zn-Co DMC catalysts, an excess of ZnCl₂ to K₃[Co(CN)₆] (e.g., molar ratio = 10 : 1) and an organic additive (tert-butanol) are usually used to produce catalysts with amorphous structures. According to the FTIR spectrum and XRD pattern of the Cu-Co DMC shown in this work, the use of excess CuCl₂ to K₃[Co(CN)₆] and tert-butanol as an organic additive showed no effect on the crystal structure of the resultant catalyst. In fact, the ratio of Cu to Co (1.7) is slightly higher than the stoichiometric ratio (1.5). In view of a sustainable and environmentally friendly approach, the authors should investigate Cu-Co PBA prepared using a stoichiometric Cu/Co ratio without using tert-butanol for the synthesis of phosphoramidates.
2. Scheme 2: ref. 21 is not relevant.
3. It would be helpful for readers if the actual or estimated molecular structure of the Cu-Co DMC was implemented.
4. The author should clarify whether the Cu-Co DMC catalysts were pre-activated before reaction.
5. The reviewer would recommend carrying out thermalgravimetric analysis of the DMC to quantify the physisorbed and chemisorbed water molecules.
6. Scheme 3: The author should explain and implement evidence to corroborate the formation of Cu(I)-Co DMC species.
7. Figure S8: adding the spectrum of Cu-Co DMC before adsorption of pyridine would help the reader to quickly evaluate the presented results.

Reviewer #2 (Remarks to the Author):

The authors reported a heterogeneous Co-Cu double metal cyanide catalyst for the oxidative coupling of amines and phosphites, which showed stable activity for several runs. This work maybe of interest to people in this field. However, I don't think this paper can be accepted at current stage.

1. Please read more carefully about Prussian blue analogues, as the description of such materials is confusing (paragraph 4).
2. For the "Oxidation of iodide by Cu-Co DMC" section, the catalyst may be soluble in HCl, have you checked this? If it is, do you think this may contribute to the higher I₂ yield after adding HCl?
3. Why the crystallinity of the catalyst changed after the reaction? An IR of the post-reaction catalyst maybe useful, which can be an additional proof to the conclusions from EXAFS. If this information is included in the supporting information file, include it in the manuscript will be better.
4. If Cu leaching has been detected after the reaction, a hot filtration experiment to check the activity of

the leached Cu is necessary.

5. As the crystalline surface of the surface can be recovered after the washing and drying processes, then it is most probably that the surface was covered by reactants, intermediated or products from the reaction. Then the conclusion “a change in the internal crystalline structure of the catalyst and the formation of two different phases during its reuse” is not appropriate.

6. Most of the references are quite old, it would be much better for the authors to check the most recent publications on this topic as well as on this kind of materials.

Reviewer #3 (Remarks to the Author):

In this work, a Cu-Co DMC catalyst is utilised as a new catalyst for phosphoramidate synthesis. It is able to be prepared readily, and can be reused for further reactions. It is a useful addition to the known DMC catalyst applications however, the reaction scope is very limited to be a highly competitive catalyst or methodology for phosphoramidate synthesis. The material itself is extensively characterised and discussed very well. However, this section appears to be emphasised too greatly to fit the title and overall aim of the work. I therefore do not recommend the manuscript is published in Comms Chem in its current state and needs major revisions to be suitable. Please consider the points below for future experiments and iterations of this work.

- Typically, the characterisation of a catalyst is discussed first and then it's catalytic output and mechanism. The authors should critically discuss what emphasis the manuscript should have as the story is not concise as currently written or fits the title. I would personally adjust the emphasis being more on the reaction scope and catalytic output/mechanism, rather than having the characterisation discussed in such detail at the end of the article because the catalyst has been described before, although I do appreciate the X-ray data is novel and interesting.
- Scheme 1 caption should be referenced appropriately.
- Synthesis of Cu-Co DMC: To help with reproducibility, what centrifugation conditions were used (time, speed etc.)?
- The reaction optimisation section should be more concise as the emphasis should be on the Cu-Co-DMC and not on previous systems. Therefore, it would be more succinct and less overwhelming to put the other Cu based catalysts from Table 1 into the SI as an extended reaction optimisation section and discuss this appropriately in the main text. I believe the focus should be on the Cu-Co-DMC as much as possible, all the other Cu catalysts have either been described previously or act as controls, therefore use these as points of discussion to accentuate the Cu-Co-DMC as a useful catalyst for these transformations.
- What is the outcome of the reaction using the optimised conditions (entry 20) without O2 balloon? How long does it take to reach the same yield as with the balloon? Control without catalyst in the optimised conditions should also be stated.

- The reaction scope needs further elaboration in order for the Cu-Co-DMC to really shown as an efficient catalyst for phosphoramidate synthesis. Can the methodology be applied to aromatic amines with various EWG/EDG substituents? What is the functional group tolerance? Could industrially relevant phosphoramidates be exemplified (nucleoside conjugates etc.)? Is the reaction scalable?
- An estimate of turnover number or any other kinetic data would be interesting.
- Are there any side products identified for lower yield reactions? If hydrolysis products are identified, what are the yields and could the reactions be run in dry solvents to see if this is minimised?
- NMR spectra of prepared compounds should be present in the SI.

Response to reviewers:

Reviewer #1:

In the preparation of Zn-Co DMC catalysts, an excess of ZnCl₂ to K₃[Co(CN)₆] (e.g., molar ratio = 10 : 1) and an organic additive (*tert*-butanol) are usually used to produce catalysts with amorphous structures. According to the FTIR spectrum and XRD pattern of the Cu-Co DMC shown in this work, the use of excess CuCl₂ to K₃[Co(CN)₆] and *tert*-butanol as an organic additive showed no effect on the crystal structure of the resultant catalyst. In fact, the ratio of Cu to Co (1.7) is slightly higher than the stoichiometric ratio (1.5). In view of a sustainable and environmentally friendly approach, the authors should investigate Cu-Co PBA prepared using a stoichiometric Cu/Co ratio without using *tert*-butanol for the synthesis of phosphoramidates.

Considering the reviewer's comment, we have investigated the effects of using a Cu-Co DMC catalyst prepared using a stoichiometric Cu/Co ratio instead of an excess of the M¹ precursor salt (CuCl₂) and without the use of *tert*-butanol as complexing agent on the synthesis of phosphoramidates (*page 13, paragraph 2*). As can be seen in *Entry 18* of **Table 1**, this modification of the catalyst preparation resulted in a noticeable (even if slight) decrease of the performance of the catalyst in the reaction, highlighting the importance of employing the original preparation conditions.

2. Scheme 2: ref. 21 is not relevant.

We have corrected Scheme's 2 caption (*page 11, Scheme 2*) by removing *ref. 21* and adding a relevant reference (9 - Tan, C. *et al.* Practical synthesis of phosphinic amides/phosphoramidates through catalytic oxidative coupling of amines and P(O)-H compounds. *Chem. Eur. J.* 26, 881-887 (2020).

3. It would be helpful for readers if the actual or estimated molecular structure of the Cu-Co DMC was implemented.

We have added the theoretical crystal structure of the Cu-Co DMC catalyst to the manuscript (*page 4, Scheme 1*). We believe this will help the reader to visualize and better understand the solid catalyst, and we thank the reviewer for the suggestion.

4. The author should clarify whether the Cu-Co DMC catalysts were pre-activated before reaction.

As the reviewer correctly recommends, a statement clarifying whether there is a pre-activation step before the reaction has been made (*page 5, paragraph 1*). As stated in the text, no pre-treatment or pre-activation of the catalyst before reaction is performed.

5. The reviewer would recommend carrying out thermalgravimetric analysis of the DMC to quantify the physisorbed and chemisorbed water molecules.

Following the recommendation of the reviewer, a thermogravimetric analysis (TGA) of the DMC has been done. The results of this measurement are mentioned in the main text (*page 6, paragraph 1*) and the data were added to the supporting information (*page 9 SI, Figure S14*). Quantification of the physisorbed and chemisorbed water molecules resulted in 17% weight loss of the analyzed sample.

6. Scheme 3: The author should explain and implement evidence to corroborate the formation of Cu(I)-Co DMC species.

The Cu sites in our Cu-Co DMC can catalytically oxidize the iodide during the aerobic oxidative coupling of amines and phosphites, regenerating the consumed I_2 . In this process, the Cu^{2+} is reduced to Cu^+ , before being regenerated by the O_2 present in the medium. In the future, we plan a dedicated *in-situ* study of the reaction, in order to observe more clearly the expected changes in the oxidation state of the Cu ions during the reaction.

With the *ex-situ* characterization carried out so far, it remains too speculative to unequivocally claim the presence of Cu(I) species, since we expect the Cu ions to return to their +2 oxidation state after every catalytic cycle. However, some hints on the aforementioned change in the oxidation state can be found in the X-ray absorption data collected for the samples after reaction. Firstly, the Cu K-edge XANES of the spent Cu-Co DMC (after reaction) was characterized by an edge shift towards slightly lower energies (**Fig. 5**). Secondly, XAS measurements for the washed Cu-Co DMC (recovered after reaction) indicate the presence of an additional phase, resulting in a perfect reconstruction of the spectrum by a linear combination of Cu atoms coordinated in square planar, octahedral and tetrahedral manner (**Fig. 6**). For the tetrahedral coordination, Cu(I) was added as reference, which could hint at the presence of such Cu(I) species.

7. Figure S8: adding the spectrum of Cu-Co DMC before adsorption of pyridine would help the reader to quickly evaluate the presented results.

As recommended by the reviewer, the FTIR spectrum of the Cu-Co DMC before adsorption of pyridine has been added to the supporting information (*page 4 SI, Figure S6*). No absorption bands are observed in this particular region.

Reviewer #2:

1. Please read more carefully about Prussian blue analogues, as the description of such materials is confusing (paragraph 4).

Considering the reviewer's suggestion, DMC's general definition has been rephrased in a clearer way (page 3, paragraph 2). Additionally, a new relevant reference has been added to the text (13 - Valvekens, P. *New Materials for Catalytic Applications*, 1-12, Elsevier, 2016).

2. For the "Oxidation of iodide by Cu-Co DMC" section, the catalyst may be soluble in HCl, have you checked this? If it is, do you think this may contribute to the higher I₂ yield after adding HCl?

Responding to the concerns expressed by the reviewer on the stability of the catalyst under acid conditions during the oxidation of iodide in methanolic solution, we investigated the structural integrity of the Cu-Co DMC catalyst in solutions of different concentrations of HCl by XRD. The results of these experiments are mentioned and discussed in the main text (page 21, paragraph 2); the stability of the catalysts under the tested conditions is confirmed. The data was also added to the supporting information (page 8 SI, Figure S13).

3. Why the crystallinity of the catalyst changed after the reaction? An IR of the post-reaction catalyst maybe useful, which can be an additional proof to the conclusions from EXAFS. If this information is included in the supporting information file, include it in the manuscript will be better.

As the reviewer suggests, we now present the FTIR spectra in the main manuscript, **Fig. 2** (page 17). This data is then discussed in the manuscript (page 16, paragraph 1). The characteristic C≡N band of the DMC shifts to lower wavenumbers for the sample after reaction, indicating changes in the electron donation to its surrounding metals, thus explaining the change in crystallinity.

4. If Cu leaching has been detected after the reaction, a hot filtration experiment to check the activity of the leached Cu is necessary.

Following the reviewer's recommendation, a hot filtration test was carried out under the optimized reaction conditions and the results were added to the supporting information (page 5 SI, Figure S7) and discussed in the main text (page 14, paragraph 1), revealing the heterogeneous nature of the catalytic system.

5. As the crystalline surface of the surface can be recovered after the washing and drying processes, then it is most probable that the surface was covered by reactants, intermediates or

products from the reaction. Then the conclusion “a change in the internal crystalline structure of the catalyst and the formation of two different phases during its reuse” is not appropriate.

We thank the reviewer for this comment and feedback. Indeed we believe that after reaction the surface and active sites are covered by reactants (and products), and we have made that observation more clear in the manuscript (page 25, paragraph 2 and page 26, paragraph 1).

By our initial assessment using XRD and FTIR, it was not possible to obtain a deeper understanding regarding the changes in the structure of the DMC. Therefore, we were motivated to study the local structure of the metal atoms of the catalyst more in depth by XAS. These studies revealed not only that reactants, intermediates or products from the reaction might be covering the surface of the DMC, but also that the Cu atoms changed their coordination geometry, from octahedral to square planar. With this in consideration, we have modified our conclusion accordingly to reflect our findings in a more appropriate manner.

6. Most of the references are quite old, it would be much better for the authors to check the most recent publications on this topic as well as on this kind of materials.

Taking this recommendation into account, new references have been added to the manuscript: 10 - a) Wang, L., et. al., *A general copper-catalysed enantioconvergent radical Michaelis–Becker-type C(sp³)–P cross-coupling*. *Nat. Synth.* (2023) b) Gupta, S., Baranwal, S., Chaudharya, P. and Kandasamy, J., *Copper-promoted dehydrogenative cross-coupling reaction of dialkyl phosphites with sulfoximines*. *Org. Chem. Front.*, 6, 2260-2265 (2019), referring to promising results of aerobic oxidation of phosphites by Cu salts, 14 - b) Tran, C., et. al. *Organonitriles as complexing agents for the double metal cyanide-catalyzed synthesis of polyether, polyester, and polycarbonate polyols*. *Catal. Today*, 418, 114125 (2023). c) Verma, A., et al., *EDTA incorporated Fe-Zn double metal cyanide catalyst for the controlled synthesis of polyoxypropylene glycol*. *J. Polym. Res.*, 30, 62 (2023), referring to the use of different additives during the preparation of DMCs to modify their catalytic properties and 16 - g) Zhang, X., et. al. *Construction and arm evolution of trifunctional phenolic initiator-mediated polycarbonate polyols produced by using a double metal cyanide catalyst*. *Polym. Chem.*, 14, 1263-1274 (2023), referring to new applications of DMCs recently reported in the literature.

Reviewer #3:

1. Typically, the characterisation of a catalyst is discussed first and then it's catalytic output and mechanism. The authors should critically discuss what emphasis the manuscript should have as the story is not concise as currently written or fits the title. I would personally adjust the emphasis being more on the reaction scope and catalytic output/mechanism, rather than having the characterisation discussed in such detail at the end of the article because the catalyst has been described before, although I do appreciate the X-ray data is novel and interesting.

The issue was addressed by reorganizing the discussion, which now highlights the innovative aspects of the study. We thank the reviewer for this comment.

2. Scheme 1 caption should be referenced appropriately.

Following the reviewer's comment, *Scheme 1* (page 4) has now an appropriately referenced caption. Additionally, *Scheme 1* (page 4) has been redesigned.

3. Synthesis of Cu-Co DMC: To help with reproducibility, what centrifugation conditions were used (time, speed etc.)?

In response to the reviewer's comment, the centrifugation conditions for the synthesis of the Cu-Co DMC have now been added to the manuscript (page 5, paragraph 1).

4. The reaction optimisation section should be more concise as the emphasis should be on the Cu-Co-DMC and not on previous systems. Therefore, it would be more succinct and less overwhelming to put the other Cu based catalysts from Table 1 into the SI as an extended reaction optimisation section and discuss this appropriately in the main text. I believe the focus should be on the Cu-Co-DMC as much as possible, all the other Cu catalysts have either been described previously or act as controls, therefore use these as points of discussion to accentuate the Cu-Co-DMC as a useful catalyst for these transformations.

In response to the reviewer's feedback, two actions were taken: firstly, **Table 1** (page 14) was revised to include only entries about the optimization of the Cu-Co DMC catalyst; secondly, entries related to other Cu-based catalysts are now presented in the supplementary information section as an extended reaction optimization section (page 5 SI, Table S5).

5. What is the outcome of the reaction using the optimized conditions (entry 20) without O₂ balloon? How long does it take to reach the same yield as with the balloon? Control without catalyst in the optimized conditions should also be stated.

Answering the reviewer's questions, we studied the reaction under the optimized conditions in the absence of the O₂ balloon, using air as source of O₂, and performed a control reaction in the absence of catalyst under the optimized conditions. Results of these experiments are presented in *entries 17 and 19* respectively of **Table 1**, and further discussed in the main text (*page 12, paragraph 2 and 3*). In short, it was found that when no O₂ balloon is used the phosphoramidate yield decreases slightly (96% vs 99%) and the reaction takes 6 times longer to finish (3 h vs 0.5 h). A control experiment without catalyst in the optimized conditions only shows a stoichiometric yield of the product (12%) with respect to the amount of iodine added.

6. The reaction scope needs further elaboration in order for the Cu-Co-DMC to really shown as an efficient catalyst for phosphoramidate synthesis. Can the methodology be applied to aromatic amines with various EWG/EDG substituents? What is the functional group tolerance? Could industrially relevant phosphoramidates be exemplified (nucleoside conjugates etc.)? Is the reaction scalable?

We have taken the reviewer's feedback into account and further elaborated the reaction scope. We have extended the synthesis of phosphoramidates to the preparation of new compounds, employing aromatic amines (*page 20, entries 10-12 of Table 2*), amino acids (*page 20, entries 13 and 14 of Table 2*) and a cyclic phosphite (*page 20, entry 15 of Table 2*) as coupling partners. The results are discussed in the main text (*page 18, paragraph 1*).

We were however unable to find a commercially available substrate for the synthesis of nucleosides. Instead, the coupling of phosphites and the selected amino acids should show the potential application of this methodology for the synthesis of these molecules.

Finally, proof of the reaction scalability is discussed on *page 19, paragraph 1*, where a 20 g batch of dibutyl phenylethylphosphoramidate was prepared.

7. An estimate of turnover number or any other kinetic data would be interesting.

As suggested by the reviewer, we have calculated the turnover of the catalytic reaction (turnover frequency, TOF) and the result is presented in the main text (*Table 1*).

8. Are there any side products identified for lower yield reactions? If hydrolysis products are identified, what are the yields and could the reactions be run in dry solvents to see if this is minimised?

We identified the main side product of the reaction: the corresponding phosphate. This type of side product could be formed by hydrolysis of the phosphoryl halide intermediate during the reaction. We investigated the effect of the use of dry solvents on the reaction yields as suggested by the

reviewer. These results are now described in the main text (*page 17, paragraph 1*). The use of dry solvents was not very useful in suppressing the formation of this side product.

9.NMR spectra of prepared compounds should be present in the SI.

As suggested by the reviewer, NMR spectra of the crude mixture of prepared compounds has been added to the supporting information (*page 17 SI*).

REVIEWERS' COMMENTS:

Reviewer #1 (Remarks to the Author):

The authors have adequately addressed all issues mentioned in the reviewers' comments. I therefore recommend the acceptance for the revised manuscript to publish in Communications Chemistry.

Reviewer #2 (Remarks to the Author):

The authors have clarified most of the confusions, and I think this manuscript can be accepted for publication after a minor revision regarding the following problem.

1. From your hot filtration reaction, it is obvious that 1.6% Cu can account for 10% product yield. So, the solubility of this catalysts in HCl is important to evaluate the catalytic performance after adding HCl in the reaction system. Instead of XRD, I think ICP of the post reaction catalysts would be a much better way to evaluate the real active species after adding HCl.

Reviewer #3 (Remarks to the Author):

The manuscript is has greatly improved and all reviewer comments have been addressed well. I therefore recommend the article is published as is.

Response to reviewers:

Reviewer #2 (Remarks to the Author):

The authors have clarified most of the confusions, and I think this manuscript can be accepted for publication after a minor revision regarding the following problem.

1. From your hot filtration reaction, it is obvious that 1.6% Cu can account for 10% product yield. So, the solubility of this catalysts in HCl is important to evaluate the catalytic performance after adding HCl in the reaction system. Instead of XRD, I think ICP of the post reaction catalysts would be a much better way to evaluate the real active species after adding HCl.

ICP-OES analyses post reaction have been performed, which revealed that no Cu species leached during the reaction.